# BMP signaling maintains auricular chondrocyte identity and prevents microtia development by inhibiting protein kinase A

Ruichen Yang[1†], Hongshang Chu[1†], Hua Yue[2], Yuji Mishina[3], Zhenlin Zhang[2], Huijuan Liu[1*], Baojie Li[1,4*]

[1]Bio-X Institutes, Key Laboratory for the Genetics of Developmental and Neuropsychiatric Disorders, Ministry of Education, Shanghai Jiao Tong University, Shanghai, China; [2]Department of Osteoporosis and Bone Diseases, Shanghai Clinical Research Center of Bone Disease, Shanghai Jiao Tong University Affiliated Sixth People's Hospital, Shanghai, China; [3]Department of Biologic and Materials & Prosthodontics, University of Michigan School of Dentistry, Ann Arbor, United States; [4]Shanghai Institute of Stem Cell Research and Clinical Translation, Shanghai, China

**\*For correspondence:**
liuhj@sjtu.edu.cn (HL);
libj@sjtu.edu.cn (BL)

[†]These authors contributed equally to this work

**Abstract** Elastic cartilage constitutes a major component of the external ear, which functions to guide sound to the middle and inner ears. Defects in auricle development cause congenital microtia, which affects hearing and appearance in patients. Mutations in several genes have been implicated in microtia development, yet, the pathogenesis of this disorder remains incompletely understood. Here, we show that *Prrx1* genetically marks auricular chondrocytes in adult mice. Interestingly, BMP-Smad1/5/9 signaling in chondrocytes is increasingly activated from the proximal to distal segments of the ear, which is associated with a decrease in chondrocyte regenerative activity. Ablation of *Bmpr1a* in auricular chondrocytes led to chondrocyte atrophy and microtia development at the distal part. Transcriptome analysis revealed that *Bmpr1a* deficiency caused a switch from the chondrogenic program to the osteogenic program, accompanied by enhanced protein kinase A activation, likely through increased expression of *Adcy5/8*. Inhibition of PKA blocked chondrocyte-to-osteoblast transformation and microtia development. Moreover, analysis of single-cell RNA-seq of human microtia samples uncovered enriched gene expression in the PKA pathway and chondrocyte-to-osteoblast transformation process. These findings suggest that auricle cartilage is actively maintained by BMP signaling, which maintains chondrocyte identity by suppressing osteogenic differentiation.

## eLife assessment

BMP signaling plays a vital role in skeletal tissues, and the importance of its role in microtia prevention is novel and promising. This **important** study sheds light on the role of BMP signaling in preventing microtia in the ear, with **solid** data broadly supporting the claims of the authors.

## Introduction

The ear is an organ of hearing and equilibrium. The external ear (auricle), which contains many curves and folds, functions to collect and funnel sound waves into the middle and inner ear (*Anthwal and Thompson, 2016*; *Fuchs and Tucker, 2015*). The major component of the ear is elastic cartilage, a

connective tissue composed of type II collagen, proteoglycans, and glycosaminoglycans (*Bielajew et al., 2020*; *Chen et al., 2021*). Defects in auricle development can lead to microtia, a congenital disease characterized by deformed or underdeveloped ears that affects 0.83–4.34 per 10,000 births (*Gendron et al., 2016*; *Lu et al., 2020*). Microtia causes hearing problems and delays in language learning (*Gendron et al., 2016*). Genetic studies have shown that microtia is linked to mutations in *ROBO1/ROBO2*, *HOXA2*, *SIX1/SIX5*, *TBX1*, or *CHUK* in humans (*Cox et al., 2014*; *Huang et al., 2023*; *Lu et al., 2020*; *Quiat et al., 2022*). In mice, Prrx1 and Prrx2, FGF, BMP, and Wnt have been shown to affect auricle development (*ten Berge et al., 1998*; *Xie et al., 2020*). However, the molecular etiology of microtia remains incompletely understood. Currently, microtia is treated with ear reconstruction (*Evenbratt et al., 2022*; *Kobayashi et al., 2011*; *Nordberg et al., 2022*).

Cartilage is an avascular and slow-turnover tissue that receives nutrients and external signals via diffusion (*DiDomenico et al., 2018*). In addition to elastic cartilage, hyaline cartilage and fibrocartilage are found in mammals. Cartilage is located in different parts of the body and has several important functions, including support, protection, and cushioning (*Bielajew et al., 2020*; *Liu et al., 2017*). Elastic cartilage is found in the ear, epiglottis, nose, and trachea, and its main function is to provide strength and elasticity to the organs. All cartilages are formed and maintained solely by chondrocytes (*Kozhemyakina et al., 2015*). Chondrocytes are extensively studied in endochondral ossification, which forms long bones. Bone development starts with condensation of mesenchymal cells, which differentiate into chondrocytes and form a cartilage template for the bones. After proliferation, differentiation, and hypertrophic growth, hypertrophic chondrocytes undergo transformation into osteoblast/osteoprogenitors (transdifferentiation or dedifferentiation followed by redifferentiation), forming bone (*Jing et al., 2017*; *Mizuhashi et al., 2018*; *Newton et al., 2019*; *Yang et al., 2014b*; *Yang et al., 2014a*). On the other hand, auricle chondrocytes are derived from pharyngeal arches 1 and 2 during development (*Ritter and Martin, 2019*). Although the auricle contains $CD44^+CD90^+$ stem cells in the perichondrium (*Kobayashi et al., 2011*), it has limited regenerative capacity even in early postnatal mice (*Abarca-Buis and Krötzsch, 2023*).

Chondrocyte proliferation, differentiation, and hypertrophy are regulated by BMPs, FGFs, IHHs, PTHrp, and Wnts. In particular, BMP signaling positively regulates the initiation and differentiation of chondrocytes (*Murtaugh et al., 1999*; *Thielen et al., 2019*; *Ueharu and Mishina, 2023*). BMPs are members of the TGFβ superfamily, which bind to BMP receptor IA and IB and activate Smad1/5/9, which, together with Smad4, enter the nucleus to affect gene expression. *Bmpr1a* and *1b* double knockout mice have no chondrocyte formation (*Yoon et al., 2005*). Moreover, ablation of *Smad1* or *Smad5* in chondrocytes leads to severe chondrodysplasia (*Retting et al., 2009*), indicating that BMP signaling is essential for maintenance of the chondrogenic differentiation program. In addition, chondrocyte-specific knockout of *Bmpr1a* and *Bmpr1b* resulted in defects in the terminal differentiation of chondrocytes (*Yoon et al., 2006*). BMPs induce chondrocyte differentiation by controlling the expression of Sox9, which initiates and maintains the chondrocyte phenotype but inhibits chondrocyte hypertrophy (*Akiyama et al., 2002*). Sox9 may interact with Runx2 and β-catenin to regulate chondrocyte differentiation and hypertrophy (*Kozhemyakina et al., 2015*; *Takarada et al., 2013*). A recent study showed that *Sox9* is required to prevent growth plate closure postnatally, with Sox9 ablation leading to transformation from chondrocytes to osteoprogenitor cells (*Haseeb et al., 2021*).

While previous studies have uncovered essential roles for BMP signaling in chondrogenic differentiation during skeletal development, it remains unclear whether BMP signaling plays a role in cartilage maintenance in adult mice. Organ maintenance is an important process since it accounts for 90% of the lifetime of mice. Here, we provide genetic evidence that BMP signaling maintains the cell fate of auricle chondrocytes. In the absence of BMP signaling, chondrocytes become osteogenic progenitors with reduced volume, leading to shrinkage of the distal auricle and morphological changes that resemble microtia. This study uncovered a new function for BMP signaling and improved our understanding of the pathogenesis of microtia.

# Results

## *Prrx1* marks auricle chondrocytes but few epiphyseal or articular chondrocytes

We have recently shown that *Prrx1* marks CD31⁻CD45⁻CD130⁺CD200⁺CD51⁺ skeletal stem cells that are required for bone maintenance in adult mice (*Liu et al., 2022*). Our lineage tracing experiments showed that *Prrx1* marked all chondrocytes in the auricle (*Figure 1A*). This finding is in contrast to the labeling of very small populations of chondrocytes in the growth plates and articular cartilages (*Figure 1B*). As expected, *Prrx1* also marked dermal cells in the skin of the ear in adult mice (*Figure 1A*); we previously showed that *Prrx1* marks CD31⁻CD45⁻CD200⁺CD51⁺ dermal stem cells (*Liu et al., 2022*). Immunostaining results showed that auricle chondrocytes did not express Vimentin or Col1α1/2, which were mainly expressed in the dermis of the ear, or Col2 or Col10 (*Figure 1C* and data not shown). Moreover, αSMA is mainly expressed in the smooth muscle of blood vessels (*Figure 1C*). These results suggest a difference between auricle cartilage and articular cartilage and growth plates.

We also traced *Prrx1* lineage cells in *Prrx1^CreERT^; Rosa26^LSL-tdTomato^* mice that received TAM at E8.5, E13.5, or p21. We found that auricle chondrocytes were Tomato⁺ under these conditions even only one dose of TAM (1/10 of the dose for adult mice) was given to the pregnant mice at E8.5 or E13.5 (*Figure 1—figure supplement 1*). However, while E8.5 mice showed Tomato⁺ chondrocytes at both articular cartilage and growth plate, E13.5 or p21 mice showed much fewer Tomato⁺ chondrocytes at articular cartilage and growth plate (*Figure 1—figure supplement 1*). These results indicate that *Prrx1* expression differs in cartilages during development, growth, and maintenance.

## Ablation of *Bmpr1a* in *Prrx1⁺* cells in adult mice led to microtia

We generated *Prrx1^CreERT^; Bmpr1a^f/f^* mice to study the function of BMP signaling in adult cartilage. TAM was injected at 12 weeks of age. We observed a decrease in p-Smad1/5/9 on the ear sections of the *Prrx1^CreERT^; Bmpr1a^f/f^* mice 60 days after TAM administration while the protein levels were not affected (*Figure 2A* and *Figure 2—figure supplement 1A*). qPCR analysis confirmed the ablation of *Bmpr1a* in the auricle cartilage (*Figure 2B*). Note that the size and body weight of *Prrx1^CreERT^; Bmpr1a^f/f^* mice were slightly reduced (*Figure 2—figure supplement 1B*). Interestingly, the mice showed a quick decrease in the size of the external ear (*Figure 2C*), indicating a problem with auricle cartilage, which provides strength for the ear.

Histological analysis showed that the phenotype mainly occurred at the distal part of the ear. The thickness of the cartilage was slightly reduced, associated with a great reduction in the cell size (*Figure 2D*). Staining for Ki67 showed that there was no difference in the number of proliferating cells between the mutant and control mice (*Figure 2—figure supplement 1C*). Note that there were very few proliferating cells in the mutant or control mice. Staining for apoptotic cells indicated that chondrocytes did not undergo apoptosis (*Figure 2—figure supplement 1D*). Overall, these results suggest that the ear phenotype observed in *Prrx1^CreERT^; Bmpr1a^f/f^* mice is not due to decreases in the number of chondrocytes via altered proliferation and/or apoptosis; rather, it occurs through shrinkage of chondrocytes. Interestingly, deletion of *Prrx1* and its paralog gene *Prrx2* led to microtia development, while deletion of *Prrx1* leads to low-set ears and deletion of *Prrx2* does not affect ear development (*Martin et al., 1995*; *ten Berge et al., 1998*). These results suggest that these two genes play a redundant role in external ear development. However, we found no ear phenotype in *Prrx1^CreERT^* (with one *Prrx1* allele deleted) or *Bmpr1a^f/f^* mice compared to wild-type mice (*Figure 2—figure supplement 2A–C*), suggesting that ablation of one copy of *Prrx1* does not affect ear maintenance.

The growth plate and the articular cartilage appeared to be normal in *Prrx1^CreERT^; Bmpr1a^f/f^* mice (*Figure 2—figure supplement 3A and B*), likely due to the lack of labeling of chondrocytes at these locations. Moreover, Smad1/5/9 is reportedly activated in proliferating chondrocytes but not hypertrophic chondrocytes in the growth plate (*Retting et al., 2009*; *Yoon et al., 2006*).

## Chondrocytes in the distal auricle have the weakest regenerative ability

A surprising finding is that the auricle phenotype mainly occurred in the distal part (*Figure 2D*), suggesting that there exists a cellular difference along the proximal-distal axis. Our immunostaining of p-Smad1/5/9 results revealed that BMP-Smad1/5/9 signaling is more active at the distal part than at

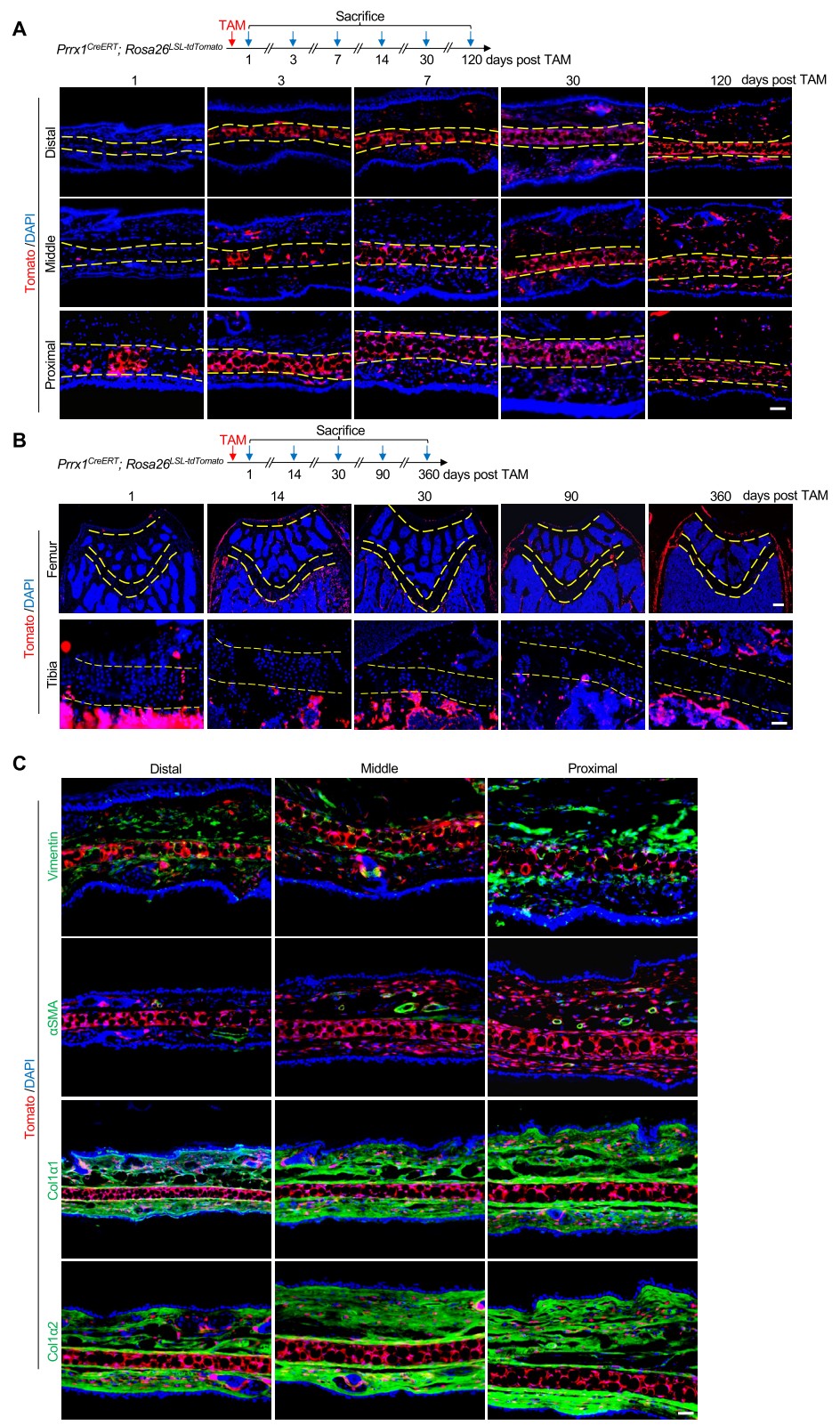

**Figure 1.** *Prrx1* marks auricle chondrocytes but few epiphyseal or articular chondrocytes. (**A**) Genetic tracing experiments showed that *Prrx1* marks all chondrocytes in the auricle. Upper panel: a diagram showing the schedule for Tamoxifen (TAM) administration and mouse euthanasia. Scale bars=50 μm. (**B**) Genetic tracing experiments showed that *Prrx1* marks few cells in the articular cartilage or the growth plates of femurs and the

*Figure 1 continued on next page*

*Figure 1 continued*

growth plates of tibias in adult mice. Scale bars (femur)=100 μm. Scale bars (tibia)=50 μm. (**C**) Representative immunostaining results for Vimentin, αSMA, Col1α1, and Col1α2 in ear sections of *Prrx1*<sup>CreERT</sup>; *Rosa26*<sup>LSL-tdTomato</sup> mice. Scale bars=50 μm.

The online version of this article includes the following figure supplement(s) for figure 1:

**Figure supplement 1.** Tracing of chondrocytes in auricle, articular cartilage, and growth plate in *Prrx1*<sup>CreERT</sup>; *Rosa26*<sup>LSL-tdTomato</sup> mice.

the proximal or middle part, in young or old mice (*Figure 2A* and *Figure 2—figure supplement 4A*). We also harvested cartilage from the proximal, mid, and distal (1/3 each) parts of normal adult mice and performed western blotting. We found that BMP-Smad1/5/9 signaling was activated to a greater extent at the distal part of the auricle (*Figure 2—figure supplement 4B*). We then tested the regenerative potential of the proximal, mid, and distal parts of the auricle by performing cartilage regeneration assays. We generated a 0.2 cm hole at these three locations and waited for 4 weeks. It was clear that the injury at the distal or middle part could not be repaired, while the injury at the proximal part was repaired (*Figure 2—figure supplement 4C and D*). These results suggest that chondrocytes at the distal part of the auricle have the lowest regenerative activity and thus may be the most differentiated, which is associated with the strongest activation of BMP-Smad1/5/9 signaling.

## Ablation of *Bmpr1a* in dermal cells did not cause microtia

Since *Prrx1* also marks dermal cells in the ear, we could not exclude the possibility that *Bmpr1a* ablation in dermal cells contributes to the ear phenotype. We found that *Col1a2* marked dermal cells but not auricle chondrocytes in *Col1a2-CreERT; Rosa26*<sup>LSL-tdTomato</sup> mice (*Figure 2—figure supplement 5A*). We generated *Col1a2-CreERT; Bmpr1a*<sup>f/f</sup> mice by crossing *Col1a2-CreERT* mice and *Bmpr1a* floxed mice. Immunostaining results showed that the levels of p-Smad1/5/9 in the dermis were greatly reduced in dermal cells in the mutant mice (*Figure 2—figure supplement 5B*). qPCR analysis confirmed the ablation of *Bmpr1a* in the dermal cells (*Figure 2—figure supplement 5C*). The ears appeared normal in *Col1a2-CreERT; Bmpr1a*<sup>f/f</sup> mice 6 months after TAM injection (*Figure 2—figure supplement 5D and E*).

In addition, we traced the *Acta2* (encoding αSMA) lineage cells in auricles and found that Acta2 could mark a small portion of chondrocytes in the middle segment of the auricle and smooth muscle cells of the blood vessels (*Figure 2—figure supplement 6A*). We also generated *Acta2-CreERT; Bmpr1a*<sup>f/f</sup> mice and found that ablation of *Bmpr1a* in *Acta2*<sup>+</sup> did not produce an ear phenotype (*Figure 2—figure supplement 6B–6E*). These findings suggest that BMP-Smad1/5/9 signaling affects ear maintenance via its effect on chondrocytes at the distal segment, with little contribution from dermal cells or smooth muscle cells.

## Ablation of *Bmpr1a* in *Prrx1*<sup>+</sup> cells in young mice also led to microtia

We then tested the effect of *Bmpr1a* ablation on *Prrx1*<sup>+</sup> in young mice. Three doses of TAM were administered to P21 *Prrx1*<sup>CreERT</sup>; *Bmpr1a*<sup>f/f</sup> mice, which were sacrificed at the age of 2 months (*Figure 3A*). We observed a decrease in p-Smad1/5/9 on the ear sections of the *Prrx1*<sup>CreERT</sup>; *Bmpr1a*<sup>f/f</sup> mice (*Figure 3A*). qPCR analysis confirmed the ablation of *Bmpr1a* in the auricle cartilage (*Figure 3B*). We found decreases in the ear size and the volumes of chondrocytes at the distal part of the ear in *Prrx1*<sup>CreERT</sup>; *Bmpr1a*<sup>f/f</sup> mice (*Figure 3C and D*). However, these mice showed normal growth plates and articular cartilage (*Figure 3—figure supplement 1A and B*), likely attributable to the lack of labeling of chondrocytes in the growth plate and articular cartilage in *Prrx1*<sup>CreERT</sup>; *Rosa26*<sup>LSL-tdTomato</sup> mice (*Figure 1—figure supplement 1*). Overall, these results suggest that BMP signaling is required for both growth and maintenance of the auricle.

## Alteration of transcription profiles in *Bmpr1a*-deficient chondrocytes

We then harvested the distal part of the auricle from three *Prrx1*<sup>CreERT</sup>; *Bmpr1a*<sup>f/f</sup> and age- and gender-matched control mice. We prepared the total RNA and performed bulk RNA-seq. We found that the transcription of at least 500 genes was affected by *Bmpr1a* ablation (*Figure 4A*). KEGG and GO analyses revealed that *Bmpr1a* ablation led to increased expression of genes in inflammation,

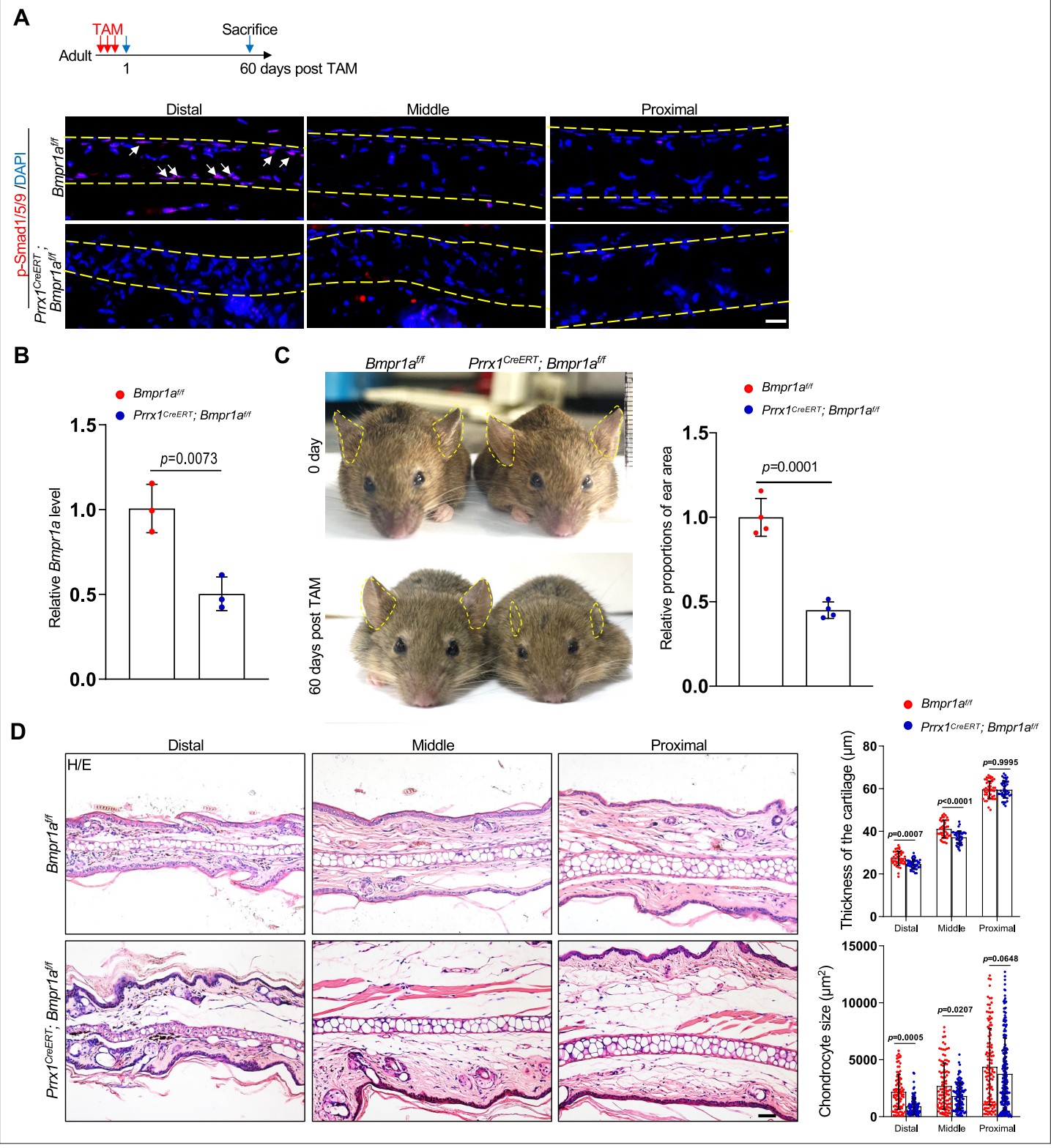

**Figure 2.** Ablation of *Bmpr1a* in *Prrx1+* in adult mice led to microtia. (**A**) A decrease in p-Smad1/5/9 on the ear sections of the *Prrx1^CreERT^; Bmpr1a^f/f^* mice. Upper panel: Schedule for Tamoxifen (TAM) administration and mouse euthanasia. Scale bars=20 µm. Arrows: positive signals. Smad1/5/9 staining results. (**B**) qPCR analysis of *Bmpr1a* mRNA in the ear samples of the *Prrx1^CreERT^; Bmpr1a^f/f^* and control mice. n=3. (**C**) The ear phenotypes of the *Prrx1^CreERT^; Bmpr1a^f/f^* mice 2 months after TAM administration to adult male mice. Right panel: quantitation data. n=4. (**D**) H/E staining of ear sections from the mutant and control mice. Right panels: The thickness of the cartilage and the size of the chondrocytes in the ears of the mutant and control

*Figure 2 continued on next page*

*Figure 2 continued*

mice. Scale bars=50 µm. n=3. Unpaired two-tailed Student's t-test were applied to evaluate the correlation data in (**B and C**). Two-way ANOVA (or mixed model) multiple comparisons were applied to evaluate the correlation data in (**D**), p<0.05 was considered as statistically significant.

The online version of this article includes the following source data and figure supplement(s) for figure 2:

**Figure supplement 1.** The body weight and cell proliferation and apoptosis in the auricle of *Prrx1^CreERT^; Bmpr1a^f/f^* mice.

**Figure 2 - Figure supplement 1-source data 1**

**Figure supplement 2.** No ear phenotype in *Prrx1^CreERT^* or *Bmpr1a^f/f^* mice.

**Figure supplement 3.** Normal growth plate and articular cartilage in adult *Prrx1^CreERT^; Bmpr1a^f/f^* mice receiving Tamoxifen (TAM).

**Figure supplement 4.** Differences in the regenerative activity of the proximal, middle, and distal auricle.

**Figure 2-figure supplement 4-source data 1**

**Figure supplement 5.** Ablation of *Bmpr1a* in dermal cells does not cause microtia.

**Figure supplement 6.** Ablation of *Bmpr1a* in smooth muscle cells does not cause microtia.

stemness, cell cycle, mesenchymal cell development, proximal/distal patterning, ear morphogenesis, wound healing, TGFβ signaling, and PKA signaling, whereas genes in chondrocyte differentiation, stem cell division, skeletal morphogenesis, extracellular matrix (ECM) genes, cartilage development, Hedgehog signaling, and Wnt signaling were suppressed (*Figure 4B and C*). These transcriptome results suggest that *Bmpr1a* ablation may cause chondrocyte-to-osteoblast transformation.

Further analysis showed that the expression of many ECM proteins was down-regulated in the auricle cartilage of *Prrx1^CreERT^; Bmpr1a^f/f^* mice (*Figure 4—figure supplement 1A*). Note that mRNA levels of Smad1/5/9 were unaltered in the auricle cartilage of the mutant mice (*Figure 4—figure supplement 1B*). Immunostaining revealed that the expression of aggrecan and Col10 in the growth plates was unaltered in the mutant mice (*Figure 4—figure supplement 1C*), likely due to the lack of marking of growth plate chondrocytes by *Prrx1* in adult mice. Since ECM proteins are secreted by chondrocytes, we also checked the expression of proteins involved in protein trafficking and found that some were up-regulated and some were down-regulated (*Figure 4—figure supplement 1D*). The change in the expression of ECM and protein trafficking genes may reflect the shift from chondrocytes to osteoblasts and warrants further investigation. However, the expression of ER or Golgi stress-related genes, which play critical roles in chondrocyte differentiation and survival (*Horigome et al., 2020*; *Wang et al., 2018*), was not altered by *Bmpr1a* ablation (*Figure 4—figure supplement 1E and F*).

### *Bmpr1a* deficiency leads to an auricle chondrocyte switch to osteoblasts

The heatmaps of osteoblast-related and chondrocyte-related genes further support a chondrocyte-to-osteoblast transformation in the absence of BMP signaling (*Figure 5A* and *Figure 5—figure supplement 1A*). To validate the above findings, we stained the auricle for Col1α1, a marker for osteoblasts, and found that Col1α1 expression was elevated in the auricle of the *Prrx1^CreERT^; Bmpr1a^f/f^* mice compared to the control mice (*Figure 5B*). We also stained for alkaline phosphatase (ALP), a marker for osteoblasts, and found that ALP activity was greatly increased in the mutant mice (*Figure 5C*). WB confirmed the increases in the expression of osteoblast marker Runx2 and osteocalcin (*Figure 5D*). These results suggest that in the absence of BMP signaling, auricle chondrocytes transformed into osteoblasts/osteoprogenitors via transdifferentiation or dedifferentiation/redifferentiation. Previous studies have shown that Sox9 is involved in chondrocyte to osteoblast transformation (*Lefebvre et al., 2019*). However, we found that *Sox9* mRNA levels were not altered by *Bmpr1a* deficiency (*Figure 5—figure supplement 1B and C*).

### *Bmpr1a* deficiency led to hyperactivation of the PKA pathway

To understand the molecular mechanisms underlying the development of microtia, we examined the signaling pathways altered by *Bmpr1a* deficiency in auricle samples. We focused on the enhanced activation of the protein kinase A pathway, which is known to regulate chondrocyte differentiation. PKA signaling has been shown to promote chondrocyte differentiation but inhibit chondrocyte hypertrophy (*Dy et al., 2012*; *Tsang and Cheah, 2019*). In addition, PKA-phosphorylated Sox9 (*Huang*

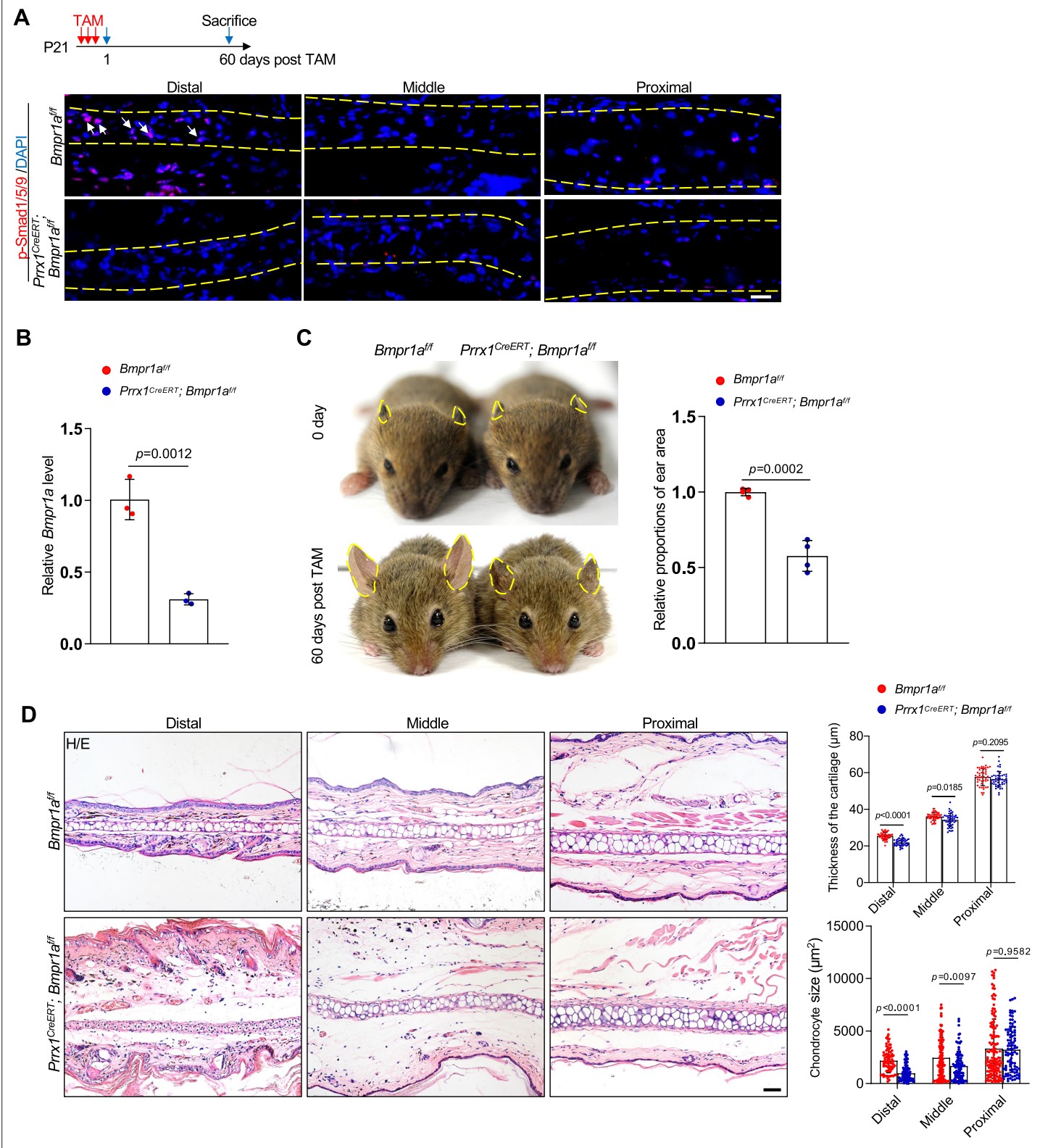

**Figure 3.** Ablation of *Bmpr1a* in *Prrx1*⁺ in young mice led to microtia. (**A**) A decrease in p-Smad1/5/9 on the ear sections of the *Prrx1*^CreERT^; *Bmpr1a*^f/f^ mice that received Tamoxifen (TAM) at P21. Upper panel: Schedule for TAM administration and mouse euthanasia. Scale bars=20 μm. Arrows: positive signals. (**B**) qPCR analysis of *Bmpr1a* mRNA in the ear samples of the *Prrx1*^CreERT^; *Bmpr1a*^f/f^ and control mice. n=3. (**C**) The ear phenotypes of the adult *Prrx1*^CreERT^; *Bmpr1a*^f/f^ mice that received TAM at P21. Right panel: quantitation data. n=4. (**D**) H/E staining of ear sections from the mutant and control mice. Right

*Figure 3 continued on next page*

*Figure 3 continued*

panels: The thickness of the cartilage and the size of the chondrocytes in the ears of the mutant and control mice. Scale bars=50 μm. n=3. Unpaired two-tailed Student's t-test were applied to evaluate the correlation data in (**B and C**). Two-way ANOVA (or mixed model) multiple comparisons were applied to evaluate the correlation data in (**D**), p<0.05 was considered as statistically significant.

The online version of this article includes the following figure supplement(s) for figure 3:

**Figure supplement 1.** Normal growth plate and articular cartilage in *Prrx1^CreERT*; *Bmpr1a^f/f* mice receiving Tamoxifen (TAM) at P21.

*et al., 2000*) may promote bone formation (*Siddappa et al., 2008*). Immunoassaying for p-Creb, a substrate of PKA, suggests that PKA is activated in auricle chondrocytes compared to the control counterparts (*Figure 6A*). Western blot analysis of the auricle samples confirmed this finding (*Figure 6B*).

What caused the enhanced activation of the PKA pathway? Our RNA-seq data showed that the expression of Adcy5 and Adcy8 enzymes responsible for cAMP production, was upregulated in the *Bmpr1a*-deficient auricle samples. Our qPCR analysis of auricle RNA showed that *Bmpr1a*-deficient auricle samples displayed increases in *Adcy5* and *Adcy8* (*Figure 6C*).

## Inhibition of PKA signaling blocks microtia development in *Bmpr1a*-deficient mice

To demonstrate the function of elevated PKA signaling in the development of microtia, we administered the PKA inhibitor H89 to *Prrx1^CreERT*; *Bmpr1a^f/f*; and control mice immediately after TAM injection (*Figure 7A*). After 2 months, we found that p-Creb was greatly suppressed and that the microtia phenotype was rescued in the mutant mice by H89, although the inhibitor did not affect the auricle in the normal mice (*Figure 7B and C*). Histological analysis revealed that chondrocyte atrophy and the decrease in ear size and cartilage thickness were rescued by H89 (*Figure 7D*). Staining results revealed that the levels of Col1α1 and the ALP activity returned to normal (*Figure 7E and F*). Western blot analysis of the auricle samples confirmed that inhibition of PKA rescued the abnormal Runx2 and osteocalcin expression (*Figure 7G*). Overall, these results suggest that elevated PKA signaling is responsible for microtia development and the transformation from chondrocytes to osteoblasts, consistent with the pro-osteogenic activity of PKA signaling.

## Human microtia samples show increased PKA signaling and osteogenic differentiation

The above studies uncover a critical role for the BMP-PKA pathway in controlling auricle chondrocyte identity and microtia pathogenesis in mice. Literature search did not uncover any study on *Prrx1* in auricle chondrocytes in humans. We also analyzed a single-cell RNA-seq dataset of the pinna of 2 microtia patients who may have disrupted *ROBO1/2* expression (*Quiat et al., 2022*) and detected a population of cells with osteoblast features besides chondrocytes, fibroblasts, and endothelial cells (*Figure 7—figure supplement 1A–D*). Moreover, we found enriched gene expression in the PKA pathway, which was associated with *ADCY2* (*Figure 7—figure supplement 1C–1E*). These results suggest a chondrocyte-to-osteoblast transformation under the influence of the PKA pathway in microtia patient samples. Although recent studies have uncovered an interaction between the BMP pathway and the Robo pathway (*Morita et al., 2023*; *Tumelty et al., 2018*), the functional link between these pathways in chondrocytes awaits further investigation.

## Discussion

Microtia is usually a congenital disorder that affects the hearing and psychological health of affected children. Many of the disease-candidate genes are involved in the proliferation and differentiation of auricle chondrocytes and, therefore, the development or growth of auricle cartilage (*Gendron et al., 2016*; *Huang et al., 2023*). Here, we show that in young or adult mice, ablation of *Bmpr1a* in auricle chondrocytes, which are genetically labeled by *Prrx1*, leads to rapid development of microtia. The function of BMP signaling in mature auricle chondrocytes appears to maintain cell identity and prevent chondrocytes from transforming into osteoblasts/osteoprogenitors. These results suggest that auricle appearance and integrity are actively maintained by BMP signaling.

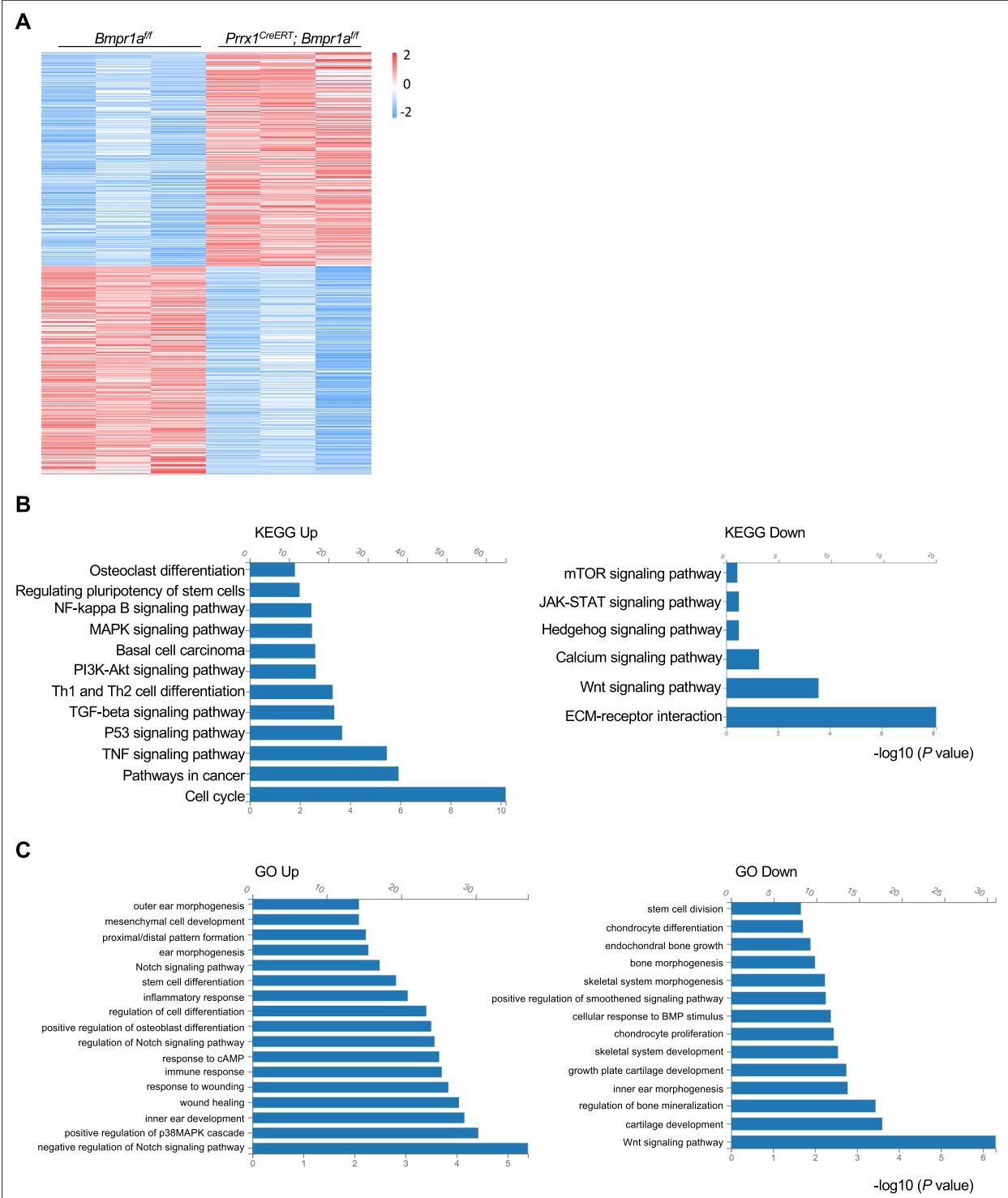

**Figure 4.** Alteration of transcription profiles in *Bmpr1a*-deficient chondrocytes. (**A**) Heatmaps of the top 500 genes expressed in the distal part of the auricle of the *Prrx1^CreERT*; *Bmpr1a^f/f* and age- and gender-matched control mice. n=3. (**B**) KEGG analysis results of the pathways affected by *Bmpr1a* ablation. (**C**) Gene Ontology (GO) analysis results of the modules affected by *Bmpr1a* ablation.

The online version of this article includes the following figure supplement(s) for figure 4:

**Figure supplement 1.** Expression of extracellular matrix (ECM) and protein trafficking-related genes in the auricle of control and *Bmpr1a*-deficient mice.

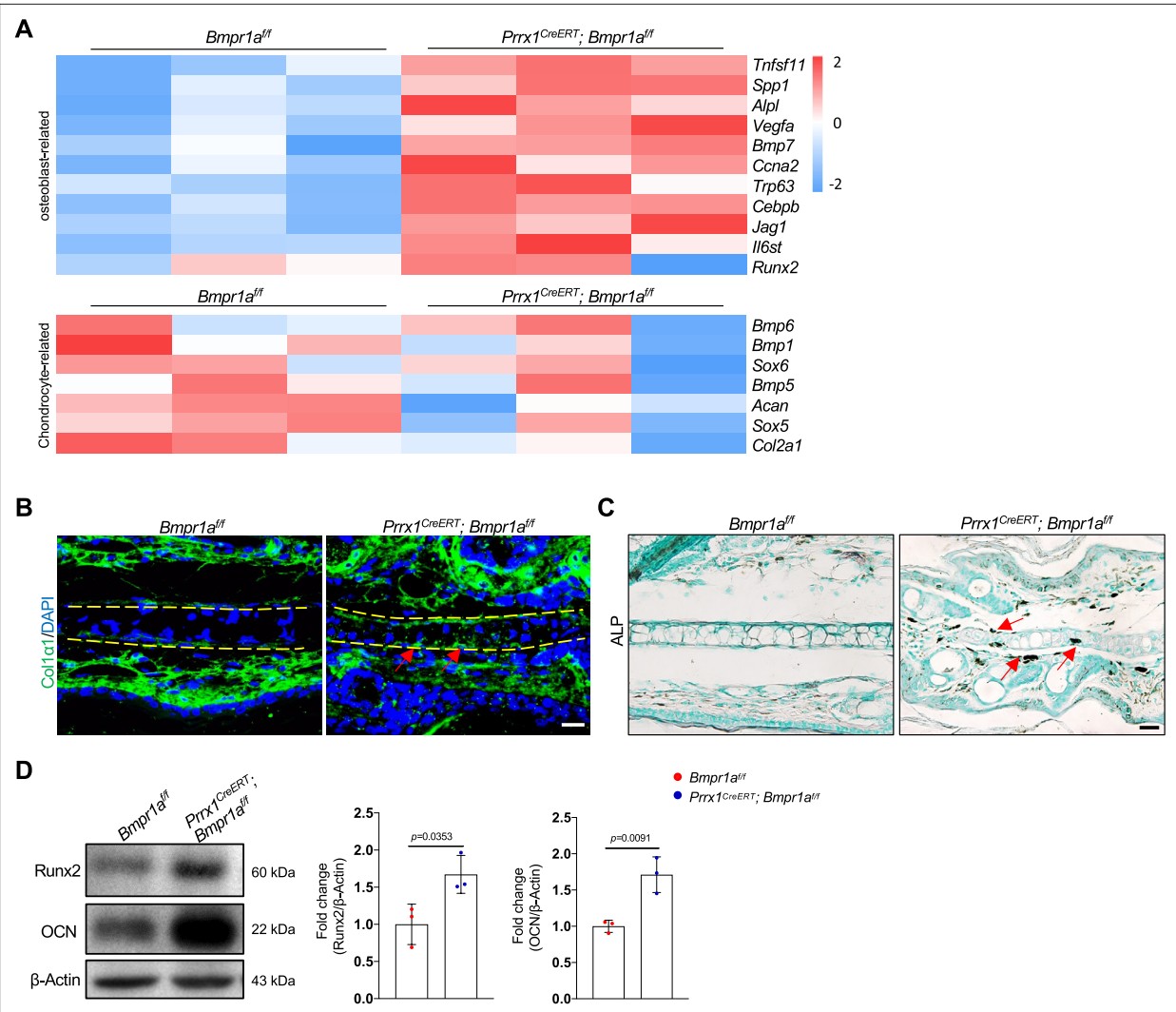

**Figure 5.** *Bmpr1a* deficiency leads to an auricle chondrocyte fate switch to osteoblasts. (**A**) Heatmaps of osteoblast-related genes and chondrocyte-related genes differentially expressed in the *Bmpr1a*-deficient and control mice. n=3. (**B**). Immunostaining results for Col1α1 in ear sections of the *Bmpr1a*-deficient and control mice. Scale bars=20 μm. Arrows: positive signals. (**C**) Alkaline phosphatase (ALP) activity was greatly increased in the auricle of the *Prrx1^CreERT; Bmpr1a^{f/f}* mice compared to the control mice. Scale bars=50 μm. Arrows: positive signals. (**D**) WB results showed that the expression of the osteoblast marker genes Runx2 and osteocalcin was increased in the mutant samples. Right panel: quantitation data. n=3. Unpaired two-tailed Student's t-test were applied to evaluate the correlation data in (**D**), p<0.05 was considered as statistically significant.

The online version of this article includes the following source data and figure supplement(s) for figure 5:

**Source data 1.** Original file for the western blot in *Figure 5D*.

**Figure supplement 1.** Expression of chondrocyte-related genes and *Sox9* in the auricle of control and *Bmpr1a*-deficient mice.

Our study thus uncovered a novel function of BMP signaling in addition to its critical roles in chondrocyte proliferation and differentiation during endochondral ossification (*Thielen et al., 2019*). Defects in BMP-Smad1/5/9 signaling lead to achondroplasia and skeletal developmental defects. On the other hand, studies of *Bmpr1a* ablation mice using *Col1-CreERT*, *Prrx1-CreERT*, or *Dmp1-Cre* lines suggest that BMP-Smad1/5/9 signaling plays a modest pro-differentiation role in osteogenic lineage cells. Instead, BMP signaling in osteogenic lineage cells regulates osteoclastogenesis by affecting the relative expression of RANKL and OPG (*He et al., 2017*; *Lim et al., 2016*; *Omi et al., 2019*; *Zhang et al., 2020*). Here, we show that BMP signaling is required to maintain the cell fate of auricle chondrocytes by preventing them from osteogenic differentiation, thus expanding the physiological roles of BMP signaling in skeletal development and maintenance (*Thielen et al., 2019*). Intriguingly, ablation of *Bmpr1a* in adult mice using a transgenic *Prrx1-CreERT* line did not produce an ear phenotype

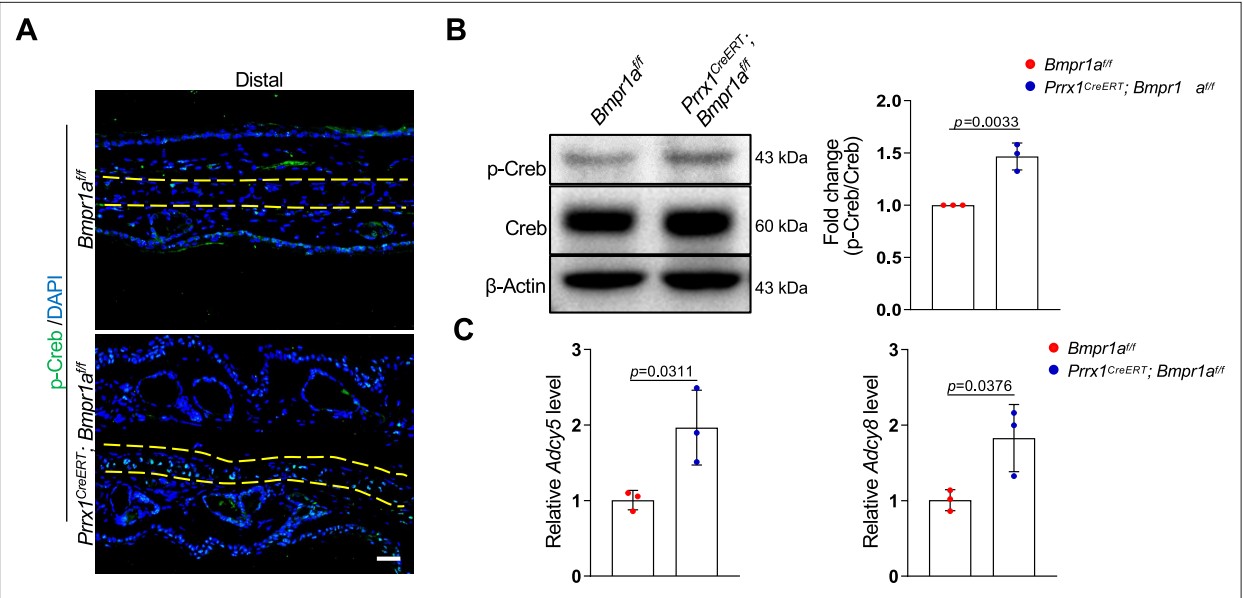

**Figure 6.** *Bmpr1a* deficiency led to hyperactivation of PKA signaling. (**A**) Immunoassaying for p-Creb in the ear sections of the *Bmpr1a*-deficient and control mice. (**B**) Western blot analysis of p-Creb in auricle samples of the *Bmpr1a*-deficient and control mice. Right panel: quantitation data. N=3. (**C**) qPCR results for *Adcy5* and *Adcy8* in auricle samples of the *Bmpr1a*-deficient and control mice. n=3. Unpaired two-tailed Student's t-test were applied to evaluate the correlation data in (**B and C**), p<0.05 was considered as statistically significant.

The online version of this article includes the following source data for figure 6:

**Source data 1.** Original file for the western blot in *Figure 6B*.

(*Biswas et al., 2018*). One explanation is that the transgenic *Prrx1-CreERT* labels too few auricle chondrocytes, as it has been for endosteal and periosteal osteoblasts in adult mice, compared to the knock-in $Prrx1^{CreERT}$ line (*Liu et al., 2022*). The difference is likely caused by the fact that the transgenic *Prrx1-CreERT* was driven by a 2.3 kilobase promoter of *Prrx1* that was inserted into an unknow location in the genome, although this needs to be verified in the future. Moreover, although we show that *Prrx1* specifically marks auricle chondrocytes but not growth plate or articular chondrocytes in adult mice, its use in auricle cartilage studies as a genetic marker is complicated by the labeling of dermal cells in the external ear.

Interestingly, we show that activation of BMP-Smad1/5/9 signaling in the auricle is uneven, with the distal part displaying the strongest activation in young or old mice. Accordingly, ablation of *Bmpr1a* in auricular chondrocytes mainly causes atrophy of the distal part. One possible explanation is that the distal part contains the most mature chondrocytes with the weakest regenerative ability. If so, it may mirror the trans-differentiation of terminally differentiated hypertrophic chondrocytes in the growth plate to osteoblasts (*Tsang and Cheah, 2019*). BMPs may be provided by chondrocytes themselves or by neighboring dermal cells, which warrants further investigation.

Our study suggests that the crosstalk between the BMP and PKA pathways plays a role in maintaining auricle chondrocyte fate. In the absence of *Bmpr1a*, the fate of chondrocytes is switched into osteoblast/osteoprogenitors, which can be blocked by PKA inhibition. Previous studies have shown that pharmacological inhibition of PKA efficiently blocks chondrogenic differentiation (*Kozhemyakina et al., 2015*). On the other hand, Sox9 inhibits chondrocyte hypertrophy and promotes the cell fate switch to osteoblasts/osteoprogenitors (*Dy et al., 2012*; *Tsang and Cheah, 2019*). This finding is consistent with a previous study showing that PKA promotes osteoblastogenesis and bone formation (*Siddappa et al., 2008*). How does BMP signaling regulate PKA activation? We found that the link is likely to be *Adcy5/8*, adenylate cyclases, whose expression is upregulated in the absence of BMP signaling. BMP signaling has been shown to suppress the expression of hepcidin and Robo4 (*Morita et al., 2023*; *Wang et al., 2017*). Since the Smad1/5/9 binding site in the promoters is very short, it may collaborate with other transcription factors to activate the transcription of target genes. In addition, BMP activates noncanonical signaling pathways, including MAPKs. As such, the underlying

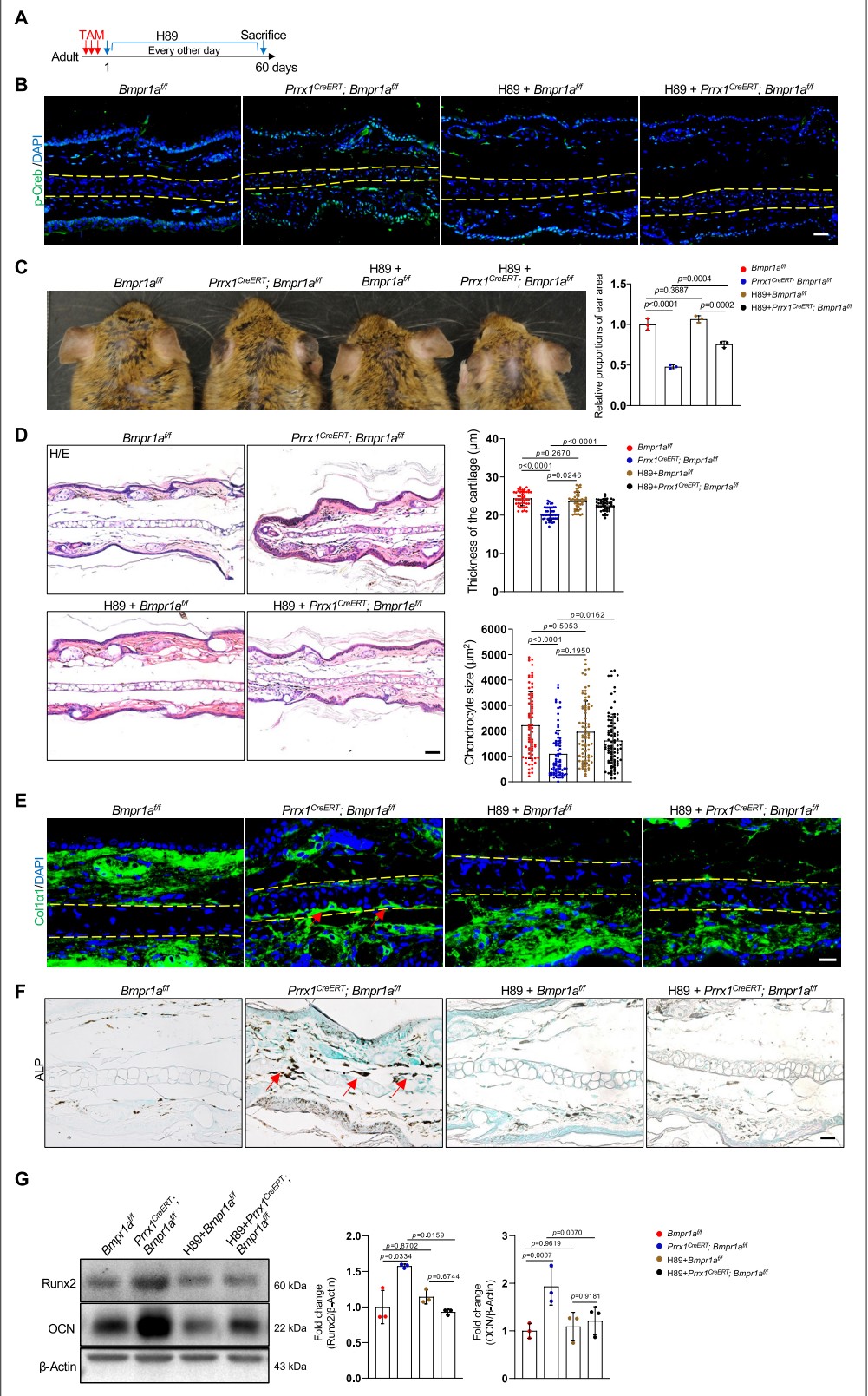

**Figure 7.** Inhibition of PKA signaling blocks microtia development in *Bmpr1a*-deficient mice. (**A**) Diagram showing the schedule of Tamoxifen (TAM) and H89 administration. (**B**) Immunostaining results of p-Creb in ear sections of the H89-treated mutant and control mice. Scale bars=50 μm. (**C**) The rescue of the microtia phenotype in the mutant mice by H89. Right panels: Quantitation data. n=3. (**D**) H/E staining results. Right panels: the thickness

*Figure 7 continued on next page*

*Figure 7 continued*

of the cartilage and the size of the chondrocytes in the ear of the mutant and control mice. n=3. Right panels: Quantitation data. Scale bars=50 μm. n=3. (**E**) Immunostaining results for Col1α1 in the ear sections of the H89-treated mutant and control mice. Scale bars=20 μm. Arrows: positive signals. (**F**) Alkaline phosphatase (ALP) activity was greatly suppressed by H89 in the auricle of the *Prrx1*$^{CreERT}$; *Bmpr1a*$^{f/f}$ mice compared to the control mice. Scale bars=50 μm. Arrows: positive signals. G. WB blot also showed that H89 rescued the expression of the osteoblast marker genes Runx2 and osteocalcin in the mutant mice. Right panel: quantitation data. n=3. One-way ANOVA (and nonparametric or mixed) multiple comparisons were applied to evaluate the correlation data in (**C, D, and G**). p<0.05 was considered as statistically significant.

The online version of this article includes the following source data and figure supplement(s) for figure 7:

**Source data 1.** Original file for the western blot in *Figure 7G*.

**Figure supplement 1.** Analysis of scRNA-seq data of human microtia pinna samples.

mechanisms for BMP-regulated expression of hepcidin, Robo4, Adcy5, and Adcy8 await further investigation (*Morita et al., 2023*; *Wang et al., 2017*).

Our findings may help to understand the pathogenesis of microtia in humans. A recent study uncovered a minimal microtia locus that carries a polymorphic repeat element, which may disrupt *ROBO1* and *ROBO2* gene expression in the auricle and be responsible for the high incidence of microtia in East Asian and Amerindian populations (*Quiat et al., 2022*). Analysis of the scRNA-seq data from microtia patients revealed that the microtia patient samples also showed enriched expression of osteoblast-related genes and the PKA pathway genes, suggesting that human microtia may share a pathogenesis similar to that in *Prrx1*$^{CreERT}$; *Bmpr1a*$^{f/f}$ mice, although this needs to be verified in the future.

In summary, this study shows that auricular chondrocytes can be genetically marked by *Prrx1*. Auricular chondrocytes, especially at the distal part of the ear, display BMP-Smd1/5/9 activation, which is required to maintain chondrocyte fate. In the absence of BMP signaling, transformation of auricular chondrocytes into osteoblasts/osteoprogenitor cells will occur, leading to microtia development. This study thus deepens our understanding of microtia pathogenesis and expands the physiological functions of BMP-PKA crosstalk.

# Materials and methods

## Key resources table

| Reagent type (species) or resource | Designation | Source or reference | Identifiers | Additional information |
|---|---|---|---|---|
| Strain, strain background (*Mus musculus*) | Prrx1$^{CreERT2}$ | Shanghai Biomodel Organism Science & Technology Development Co., Ltd | | (8 weeks to 5 months of age) |
| Strain, strain background (*Mus musculus*) | Acta2-CreERT$^2$ | The Jackson Laboratory | Strain #:032758 RRID:IMSR_JAX:032758 | (8 weeks to 5 months of age) |
| Strain, strain background (*Mus musculus*) | Col1a2-CreERT$^2$ | The Jackson Laboratory | Strain #:029567 RRID:IMSR_JAX:029567 | (8 weeks to 5 months of age) |
| Strain, strain background (*Mus musculus*) | Rosa26$^{LSL-tdTomato}$ | The Jackson Laboratory | Strain #:007909 RRID:IMSR_JAX:007909 | (8 weeks to 5 months of age) |
| Strain, strain background (*Mus musculus*) | Bmpr1a flox | Yuji Mishina's Lab | | (8 weeks to 5 months of age) |
| Antibody | anti-Ki67 (Rabbit polyclonal) | Abcam | Cat# ab15580, RRID: AB_443209 | IF(1:200) |
| Antibody | anti-Vimentin (Rabbit monoclonal) | Abcam | Cat#: ab92547; RRID:AB_10562134 | IF(1:500) |
| Antibody | anti-Col1α1 (Rabbit polyclonal) | Abcam | Cat#: ab21286; RRID:AB_446161 | IF(1:100) |

*Continued on next page*

*Continued*

| Reagent type (species) or resource | Designation | Source or reference | Identifiers | Additional information |
|---|---|---|---|---|
| Antibody | anti-Col1α2 (Mouse monoclonal) | Santa Cruz | Cat#: sc-393573; RRID:AB_2716872 | IF(1:100) |
| Antibody | anti-phospho-CREB (Rabbit polyclonal) | UPSTATE | Cat#: 06–519; RRID:AB_9310153 | IF(1:100) |
| Antibody | Anti-phospho-Smad1/5/9 (Rabbit polyclonal) | Millopore | Cat#: AB3848-I | IF(1:100) WB(1:1000) |
| Antibody | anti-RUNX2 (Rabbit monoclonal) | Cell Signaling Technology | Cat#: 12556; RRID:AB_2732805 | WB(1:1000) |
| Antibody | anti-Osteocalcin (Rabbit polyclonal) | Abcam | Cat#: ab93876; RRID:AB_10675660 | WB(1:1000) |
| Commercial assay or kit | TRACP & ALP double-stain kit | TaKaRa | Cat# MK300 | Used to label Osteoclast or Osteoblast |
| Commercial assay or kit | In Situ Cell Death Detection Kit | Roche | REF11684795910 | Used to label apoptotic cells |
| Chemical compound, drug | H89 | Selleck | S1582 | 10 mg/kg |

## Experimental design

The study was designed to examine the physiological function of BMP signaling in the maintenance of auricle chondrocyte and to elucidate the underlying molecular mechanisms.

## Mouse maintenance

The knock-in *Prrx1^CreERT2* mouse line was generated by Shanghai Biomodel Organism Science & Technology Development Co., Ltd. In this study, heterozygous *Prrx1^CreERT2* mice were used for lineage tracing or gene ablation. *Rosa26^LSL-tdTomato*, *Acta2-CreERT2*, and *Col1a2-CreERT2* mice were purchased from the Jackson Laboratory (https://www.jax.org/cn/). *Bmpr1a^f/f* mice were generated in Yuji Mishina's laboratory. Animal experiments were carried out in accordance with recommendations in the National Research Council Guide for Care and Use of Laboratory Animals and in comply with relevant ethical regulations for animal testing and research, with the protocols approved by the Institutional Animal Care and Use Committee of Shanghai, China (SYXK(SH)2011–0112). This study follows the project that 'Studies on the mechanism of MSC self-renewal differentiation and regulation of related tissue stem cells,' which was approved by Shanghai Jiao Tong University in 2015 and 2024, the approval number is A2015027 and A2024014. Male mice were used in this study. Mice were anesthetized using 40 mg kg$^{-1}$ sodium pentobarbital by intraperitoneal injection or euthanized by carbon dioxide inhalation.

## Drug administration

Tamoxifen (TAM) (Sigma) was dissolved in corn oil and administered peritoneally to mice at 120 mg kg$^{-1}$ body weight daily for 3 days (for P21 or adult mice). For E8.5 or E13.5 embryos, 1/10 of the dose was injected into the mother mice just one time. For genetic tracing or gene ablation, mice were euthanized at different time points after tamoxifen.

For in vivo inhibition of PKA, H89 (Selleck Chemicals, USA) was dissolved in 5% DMSO, 40% PEG300, 5% Tween 80, and 50% ddH$_2$O at a final concentration of 1.05 mg/ml and was administered intraperitoneally every other day after tamoxifen injection. The dosage of H89 was 10 mg/kg (in 200 µl) each time. Mice were treated for 2 months and then euthanized.

## Auricle regenerative experiments

To investigate the regenerating ability of different parts of the mouse auricle, we drilled a 2 mm (in diameter) hole at the distal, middle, or proximal parts of adult mice with a uniform aperture. The mice were euthanized 1 month later, and the ear was analyzed to assess the healing of the puncture wounds.

## RNA isolation and quantitative PCR

Chondrocyte differentiation was also assessed with quantitative PCR. Total RNA was extracted from the elastic cartilage using TRIzol reagent (Invitrogen), which was reverse transcribed using the PrimerScriptTM RT reagent Kit (TaKaRa, RR037A) to obtain cDNA. Quantitative PCR was performed using the Roche Light Cycler 480II Assay system (Roche). The levels of different mRNA species were calculated with the delta-delta CT method and normalized to *Actb*. The primer sequences are: *Bmpr1a*, sense: 5'- ATCCGATGGCTGGTTGTGCTCA-3'; antisense: 5'- CCAAATCACGGTTGTAACGACCC-3'; *Adcy5*, sense: 5'- TCGCAATGCCTACCTCAAGGAG-3'; antisense: 5'- GCGGATTGTGTCCGATGGAG TT-3'; *Adcy8*, sense: 5'-CTGCTCACAGAGACCATCTACG-3'; antisense: 5'-CAGCAGTGATGCTTCC TTGGTC-3'; *Sox9*, sense: 5'-AGTCCCAGCGAACGCACATCA-3'; antisense: 5'- GTCGTATTGCGA GCGGGTGAT-3'; *Actb*, sense: 5'-GGCTGTATTCCCCTCCATCG-3'; antisense: 5'-CCAGTTGGTAAC AATGCC ATGT-3'.

## Immunohistochemistry and immunofluorescence staining

The harvested ear tissues were fixed in 4% (vol/vol) neutral buffered formalin for 24 hr or directly embedded in optimal cutting temperature compound (OCT) by liquid nitrogen without fixation. The harvested bone tissues were fixed in 4% (vol/vol) neutral buffered formalin for 24 hr and decalcified in neutral 10% (wt/vol) EDTA solution for 1 month at room temperature on the oscillator. For paraffin sectioning, the samples were dehydrated, cleared, and embedded in paraffin blocks sequentially. The paraffin-embedded tissues were cut at 10 μm with a paraffin microtome, and the frozen-embedded tissues were cut at 8 μm with a cryostat. The ear tissue sections were incubated overnight with anti-phospho-Smad1/5/9 antibody (Millipore, AB3848-I, rabbit), anti-Ki67 antibody (Abcam, ab15580, rabbit), anti-Vimentin antibody (Abcam, ab92547, rabbit), anti-αSMA antibody (Abcam, ab5694, rabbit), anti-Col1α1 antibody (Abcam, ab21286, rabbit), anti-Col1α2 antibody (Santa Cruz, sc-393573, mouse), anti-Aggrecan antibody (Millipore, AB1031, rabbit), anti-Col10 antibody (Abcam,ab58632, rabbit), or anti-phospho-CREB antibody (UPSTATE, 06–519, rabbit). The sections were then incubated with goat anti-rabbit, mouse, or rat secondary antibodies conjugated with Alexa Fluor 488 or 555 (Thermo Fisher Scientific). Slides were mounted with antifade mounting medium and DAPI (Thermo Fisher Scientific). Images were taken under an Olympus DP72 microscope (Olympus Microsystems).

## H/E staining, ALP staining, and TUNEL staining

Paraffin sections were used for H/E staining and ALP staining with a commercial staining kit (TaKaRa, TRACP & ALP double-stain kit, Cat# MK300), while frozen sections were used for TUNEL staining with a kit (Roche, In Situ Cell Death Detection Kit, Fluorescein, REF11684795910).

## Western blotting

Western blotting was performed with standard procedures. Proteins were extracted from the distal, mid, and proximal parts of the cartilage of the mouse ear using RIPA lysis buffer with 1 mM PMSF, 1 mg/ml pepstatin, leupeptin, and aprotonin. Samples were separated on 10% SDS–PAGE gels and transferred onto PVDF membranes (Millipore). Membranes were blocked with 5% milk or 3% BSA for 1 hr and incubated with specific antibodies overnight at 4 °C on a shaker. The primary antibodies included anti-phospho-Smad1/5/9 antibody (Millipore, AB3848-I, rabbit), anti-Smad1 antibody (CST, 9743, rabbit), anti-RUNX2 antibody (CST, 12556, rabbit), anti-Osteocalcin antibody (Abcam, ab93876, rabbit), anti-β-Actin antibody (Santa Cruz, sc47778, mouse), anti-phospho-CREB antibody (UPSTATE, 06–519, rabbit), or anti-CREB antibody (UPSTATE, 04–767, rabbit). The secondary antibodies included anti-rabbit IgG, HRP-linked antibody (CST, 7074, rabbit) and anti-mouse IgG, HRP-linked antibody (CST, 7076, mouse). The WB bands were visualized using the FluoChem M system (ProteinSimple). To quantify the western blotting results, we measured the density of each band by Image J, which was normalized to the loading control of the sample.

## RNA sequencing

The distal part of the auricle of the mouse ear was harvested from control or knockout mice (n=3 per group), from which RNA was isolated. RNA-seq was conducted by BGI Genomics Co., Ltd with the Illumina system. For data processing, SOAP nuke (v1.5.2) was applied to filter reads and saved in FASTQ format. After quality control, clean reads were aligned to the reference mouse genome using

Bowtie2 (v2.2.5) with default parameters. Gene expression levels were calculated and blasted with HISAT (v2.0.4) based on the Burrows-Wheeler transform and Ferragina-Manzini. Clean reads were then detected in terms of randomicity, coverage, and degree of saturation. Statistically significant (adjusted p-value ≤0.05) genes with large expression changes (fold change ≥1.5) were defined as differentially expressed genes. In pathway analysis and Gene Ontology (GO) analysis, all candidate genes were assorted according to official classification with the KEGG or GO annotation result, and phyper (a function of R) was performed for GO and pathway functional enrichment. Furthermore, GO analysis of the stage-specific gene signature was performed in the biological process category. The p-value calculating formula was used based on previous studies (*Ge et al., 2021*), and the false discovery rate (FDR) was calculated for each p-value with FDR ≤0.05 defined as significantly enriched.

## Statistical analysis

All measurements were collected by two authors who were blinded to the allocations of the mice. Image-Pro Plus was used to measure elastic cartilage thickness, cell volume, and ear size. GraphPad Prism 9.5.0 software was used to analyze and plot the ear thickness, cell volume, ear size, qPCR data, and immunofluorescence results. All quantitative data are presented as the mean ± SD unless indicated otherwise. One-way ANOVA (and nonparametric or mixed) and two-way ANOVA (or mixed model) for multiple comparisons and unpaired two-tailed Student's t-test were applied to evaluate the correlation data, and $p<0.05$ was considered statistically significant. We ensure that we have not set criteria and exclusions in this study. For each analysis, we reported the exact value of *n* in each experimental group.

## Acknowledgements

We thank Shoutao Qiu and Yuping Li for their technical assistance.

## Additional information

### Competing interests

Yuji Mishina: Reviewing editor, *eLife*. The other authors declare that no competing interests exist.

### Funding

| Funder | Grant reference number | Author |
| --- | --- | --- |
| National Key Research and Development Program of China | 2018YFA0800803 | Baojie Li |
| National Natural Science Foundation of China | 92268101 | Baojie Li |
| National Natural Science Foundation of China | 32230045 | Baojie Li |
| Peak Disciplines (Type IV) of Institutions of Higher Learning in Shanghai | | Baojie Li |

The funders had no role in study design, data collection and interpretation, or the decision to submit the work for publication.

### Author contributions

Ruichen Yang, Data curation, Formal analysis, Validation, Visualization, Writing – review and editing; Hongshang Chu, Data curation, Formal analysis, Validation, Visualization, Methodology, Writing – review and editing; Hua Yue, Data curation, Validation; Yuji Mishina, Resources; Zhenlin Zhang, Huijuan Liu, Conceptualization, Writing - original draft, Project administration, Writing – review and editing; Baojie Li, Conceptualization, Resources, Supervision, Funding acquisition, Investigation, Writing - original draft, Project administration, Writing – review and editing

## Author ORCIDs
Ruichen Yang ⓘ http://orcid.org/0000-0003-0981-0520
Yuji Mishina ⓘ http://orcid.org/0000-0002-6268-4204
Baojie Li ⓘ http://orcid.org/0000-0002-3913-1062

## Ethics

Animal experiments were carried out in accordance with recommendations in the National Research Council Guide for Care and Use of Laboratory Animals and in comply with relevant ethical regulations for animal testing and research, with the protocols approved by the Institutional Animal Care and Use Committee of Shanghai, China (SYXK(SH)2011-0112). This study follows the project that "Studies on the mechanism of MSC self-renewal differentiation and regulation of related tissue stem cells", which was approved by Shanghai Jiao Tong University in 2015 and the approval number is A2015027. Mice were anesthetized using 40 mg kg-1 sodium pentobarbital by intraperitoneal injection or euthanized by carbon dioxide inhalation.

Reviewer #1 (Public Review): https://doi.org/10.7554/eLife.91883.3.sa1
Reviewer #2 (Public Review): https://doi.org/10.7554/eLife.91883.3.sa2
Author response https://doi.org/10.7554/eLife.91883.3.sa3

# Additional files

## Supplementary files
- Supplementary file 1. The original statistical data of this article.
- MDAR checklist

## Data availability

All data generated or analyzed in this study are included in this article, and are available from the corresponding author on request.

The following dataset was generated:

| Author(s) | Year | Dataset title | Dataset URL | Database and Identifier |
|---|---|---|---|---|
| Yang R, Chu H | 2023 | Next Generation Sequencing Facilitates Quantitative Analysis of distal part ofthe auricle from three Prrx1-CreERT:Bmprlaf/f and age- and gender-matchedcontrol mice | https://www.ncbi.nlm.nih.gov/geo/query/acc.cgi?acc=GSE240126 | NCBI Gene Expression Omnibus, GSE240126 |

The following previously published dataset was used:

| Author(s) | Year | Dataset title | Dataset URL | Database and Identifier |
|---|---|---|---|---|
| Daniel Q, Jon S | 2022 | An ancient founder mutation located between ROBO1 and ROBO2 is responsible for increased microtia risk in Amerindigenous populations (scRNA-Seq) | https://www.ncbi.nlm.nih.gov/geo/query/acc.cgi?acc=GSE202441 | NCBI Gene Expression Omnibus, GSE202441 |

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
