## [Editor Report · eLife assessment]

BMP signaling plays a vital role in skeletal tissues, and the importance of its role in microtia prevention is novel and promising. This **important** study sheds light on the role of BMP signaling in preventing microtia in the ear, with **solid** data broadly supporting the claims of the authors.

---

## [Referee Report · Reviewer #1 (Public Review)]

Summary:

In this manuscript, Ruichen Yang et al. investigated the importance of BMP signaling in preventing microtia. Authors showed that Cre recombinase mediated deletion of Bmpr1a using skeletal stem specific Cre Prx1Cre leads to microtia in adult and young mice. In these mice, distal auricle is more affected than middle and proximal. In these Bmpr1a floxed Prx1Cre mice, auricle chondrocyte start to differentiate into osteoblasts through increase in PKA signaling. The authors showed human single-cell RNA-Seq data sets where they observed increased PKA signaling in microtia patient which resembles their animal model experiments.

Strengths:

Although the importance of BMP signaling in skeletal tissues has been previously reported, the importance of its role in microtia prevention is novel and very promising to study in detail. The authors satisfied the experimental questions by performing correct methods and explaining the results in detail.

---

## [Referee Report · Reviewer #2 (Public Review)]

The authors (Yang et al.) present a well-executed study of a mouse model of Bmpr1a focusing on microtia development and pathogenesis.

The authors report that the generation of the Bmpr1a in Prrx1+ cells in adult mice helps characterize the developmental progression of the external ear.

The authors explain how auricular chondrocytes differ from growth plates or other chondrocytes and BMP-Smd1/5/9 activation, which is required to maintain chondrocyte fate in the distal part of the ear. The authors explain with evidence how BMP signaling actively maintains auricle cartilage in the post-developmental stage.

Elegant immunofluorescence staining, excellent histology preparations and dissections, excellent microscopy, sufficient experimental sample size, and good statistical analyses support the results. The study is well grounded in extensively reviewed and cited existing literature. This report sets the stage for a comprehensive interrogation of Bmpr1a deficiency and ear defects.

---

## [Author Response]

The following is the authors’ response to the original reviews.

**Reviewer #1 (Recommendations For The Authors):**
(1) Figures 1B, S4, and S5, Tibia sections would be more informative and promising as the growth plate is flat. Otherwise, histology of the knee would be preferred.

We have added the tibia section images in Figures 1B, S4, and S5 (New Figure 1B, Figure 2-figure supplement 3A, and Figure 3-figure supplement 1A).

(2) Figure 1C, The authors performed immunostaining for vimentin, alpha-SMA, Col1a1 and Col1a2. The authors should use adjusted sections for the immunostaining for different antibodies. It would avoid region-specific variations in the size and shape of sections and the data would be more reliable. Please correct and revise.

We have provided immunostaining results using consecutive sections at the similar locations of the external ear (Figure 1C).

(3) Figure 2A and throughout the manuscript where authors performed p-smad1/5/9 fluorescent immunostaining, the authors should also show non-phospho levels of p-smad1/5/9. Please correct and revise.

We have tried different anti-Smad1/5/9 antibodies and the signals have very high background and are not presentable. We instead did a western blot on auricle samples and the results are in Figure 2-figure supplement 1A, suggesting that ablation of Bmpr1a led to loss of activation of Smad1/5/9 without affecting their expression. For different segments of external ear, we also provided WB results in Figure 2-figure supplement 4B. In addition, we added RNA-seq data regarding the Smad1,5,9 mRNA levels, which were not affected by Bmpr1a ablation (Figure 4-figure supplement 1B). Overall, these results suggest that Bmpr1a ablation does not affect the expression of Smad1/5/9.

(4) Result 2, lines 131-134, the authors mentioned in the text that they observed no ear phenotype of Prrx1CreERT or Bmpr1af/f mice compared with wild-type mice (Figures S2A and S2B). However, the figures did not show histology pictures of wild-type mice. Please correct and revise.

We have provided histological pictures of wild type mice (Figure 2-figure supplement 2C).

(5) Result 5, lines 173-174 "We generated....Bmpr1a floxed mice". How did authors generate Col1a2-CreERT; Bmpr1af/f mice by crossing Prrx1Cre-ERT and Bmpr1af/f mice? Please correct and revise.

It is a typo and has been corrected.

(6) In the previous study by Soma Biswas et al., (Scientific Reports 2018, PMID 29855498) the authors mentioned in the result section that the mice with deletion of Bmpr1a using Prx1Cre looked morphologically normal. They did not mention the ear phenotype/microtia. Please explain how this study differs from current work and what are the limitations in the discussion.

We did not observe an obvious ear phenotype in the adult transgenic Prrx1-CreERT; Bmpr1af/f mice. The reason could be that that the transgene label too few auricle chondrocytes as it has been for endosteal bones and periosteal bones in adult mice (Liu et al. Nat Genet 2022; Wilk, K. et al. Stem Cell Rep 2017; Julien A et al. J Bone Miner Res 2022). The difference is likely caused by the fact that the transgenic CreERT line was driven by a 2.3 kilobase promoter of Prrx1 that was inserted to unknow location in the genome. Since we do not carry the transgenic line any more, we cannot directly test the labelling efficiency of the transgenic line in auricle. We have discussed this point in the revised manuscript.

**Reviewer #2 (Recommendations For The Authors):**
Chondrocytes are present in many parts of the body; some components are replaced by osteoblast cells, but others stay with their morphology. These cells are in different morphological and cellular conditions throughout the body.Is there any human variant study of Prrx1 and their association with auricle chondrocytes is present?

We searched the literature and found no study on Prrx1 in auricle chondrocytes in human.

Do auricle chondrocytes have Prrx1+ through their developmental stage, and what's the expression situation of Prrx1+ at articular cartilage and growth plates throughout development? Only a small population is positive throughout the development, or they lose as they develop.

We traced Prrx1 lineage cells in Prrx1-CreERT; R26tdTomato mice that received TAM at E8.5, E13.5, or p21. We found that auricle chondrocytes were Tomato+ under these conditions even only one dose of TAM (1/10 of the dose for adult mice) was given to the pregnant mice at E8.5 or E13.5 (Figure 1-figure supplement 1). However, while E8.5 mice showed Tomato+ chondrocytes at both articular cartilage and growth plate, E13.5 or p21 mice showed much fewer Tomato+ chondrocytes at articular cartilage and growth plate (Figure 1-figure supplement 1). These results indicate that Prrx1 expression differs in cartilages during development, growth, and maintenance.

What's your rationale for studying Bmpr1a ablation at the adult stage?

Organ development and maintenance are different processes, especially for slow-turnover tissues. Organ maintenance is also important since it accounts for 90% of the lifetime of mice. While previous studies have uncovered essential roles for BMP signaling in chondrogenic differentiation during development, it remains unclear whether BMP signaling plays a role in cartilage maintenance in adult mice.

Line no 128: Chondrocytes are shirked but still have normal proliferation; what's the author's thought about it?

Sorry that we did not make it clear enough. Actually there were very few cells undergoing proliferation in auricle cartilage and Bmpr1a ablation did not alter that. We have rephrased these sentences.

Do chondrocytes have protein trafficking defects or ER/Golgi stress?

We checked the expression of proteins involved in protein trafficking and found that some were up-regulated and some were down-regulated (Figure 4-figure supplement 1D), which may reflect the shift from chondrocytes to osteoblasts and warrants further investigation. However, the expression of ER or Golgi stress-related genes, which play critical roles in chondrocyte differentiation and survival (Wang et al. 2018; Horigome et al. 2020), was not altered by Bmpr1a ablation (Figure 4-figure supplement 1E and 1F).

How many Prrx paralogs are there in the system? Are all associated with auricle chondrocytes and similar mechanisms?

There is one Prrx1 paralog, Prrx2. While Prrx1-/- mice lived for up to 24 hours after birth with low-set ears (Martin JF. Eta al. Genes Dev. 1995), Prrx2-/- mice are perfectly normal. Prx1-/-Prx2-/- double mutant mice died within an hour after birth and the pups showed no external ears (ten Berge D. et al. Development. 1998). We have added this information into the revised manuscript.

Extracellular matrix (ECM) provides cell-to-cell interaction and environment for cell growth. Does Bmpr1a ablation lead to any changes in ECM at the auricle or growth plate chondrocytes?

Our analysis showed that the expression of many ECM proteins was down-regulated in auricle cartilage of Prrx1-CreERT; Bmpr1af/f mice (Figure 4-figure supplement 1A). This may reflect the shift from chondrocytes to osteoblasts and warrants further investigation. However, immunostaining revealed that the expression of Aggrecan and Col10 in the growth plates was unaltered in adult Prrx1-CreERT; Bmpr1af/f mice compared to control mice (Figure 4-figure supplement 1C), likely due to the lack of marking of chondrocytes in growth plates.

Microtia usually develops during the first trimester of pregnancy in humans. What's your view about studying at the adult stage compared to intrauterine development?

Congenital microtia is a problem with the formation of external ear whereas microtia development in adult mice is a problem with the maintenance of the auricle chondrocytes. Organ maintenance is also an important process as it starts from 3 months of age and lasts for 90% of the lifetime of mice.

In RNA sequencing protocol, Wikipedia pages keep updating, so it is very strange to cite the Wikipedia pages. Cite a research article for it.

We have replaced this reference.

Why do the authors have a very low FDR value for this study? How does this value strengthen the study?

It was a typo that has been corrected.

It needs further validation to show that Prrx1 marked cells are a good model for auricular chondrocyte-related studies.

We show that Prrx1 marks auricle chondrocytes but few growth plate or articular chondrocytes in adult mice, suggestive its specificity. However, the use of Prrx1-CreERT line in auricle cartilage studies is complicated by the labelling of dermal cells in the external ear by Prrx1. We have discussed this point in the revised manuscript.